Manuscript prepared for Atmos. Chem. Phys.
with version 2015/04/24 7.83 Copernicus papers of the LaTeX class copernicus.cls.
Date: 18 November 2016

# Dynamic climate emulators for solar geoengineering

Douglas G. MacMartin[1] and Ben Kravitz [2]

[1]Department of Mechanical and Aerospace Engineering, Cornell University, Ithaca NY, USA and
Computing + Mathematical Sciences, California Institute of Technology, Pasadena CA, USA
[2]Atmospheric Sciences and Global Change Division, Pacific Northwest National Laboratory,
Richland, WA, USA

*Correspondence to:* D. G. MacMartin (dgm224@cornell.edu)

**Abstract.** Climate emulators trained on existing simulations can be used to project the climate effects that would result from different possible future pathways of anthropogenic forcing, without relying further on general circulation model (GCM) simulations. We extend this idea to include different amounts of solar geoengineering in addition to different pathways of greenhouse gas concentrations by training emulators from a multi-model ensemble of simulations from the Geoengineering Model Intercomparison Project (GeoMIP). The emulator is trained on the abrupt $4\times CO_2$ and a compensating solar reduction simulation (G1), and evaluated by comparing predictions against a simulated 1% per year $CO_2$ increase and a similarly smaller solar reduction (G2). We find reasonable agreement in most models for predicting changes in temperature and precipitation (including regional effects), and annual-mean Northern hemisphere sea ice extent, with the difference between simulation and prediction typically smaller than natural variability. This verifies that the linearity assumption used in constructing the emulator is sufficient for these variables over the range of forcing considered. Annual-minimum Northern hemisphere sea ice extent is less-well predicted, indicating a limit of the linearity assumption.

## 1 Introduction

Climate emulators have been used extensively to provide projections of climate changes for different anthropogenic forcing trajectories. These are trained based on a limited number of simulations with General Circulation Models (GCMs) and allow prediction of climate response for a much broader set of trajectories, trading the fidelity of a GCM simulation for computational efficiency. A similar approach could in principle be undertaken for projections of the climate effects from solar geoengineering. Various solar geoengineering approaches have been suggested for intentionally influencing Earth's radiation budget, such as the injection of aerosols into the stratosphere (see, e.g., National Academy of Sciences, 2015). It is possible that such approaches may be considered in the future for reducing some amount of climate damages. However, any climate model simulation of geoengineering necessarily corresponds to some specific scenario, such as offsetting all of the global-mean-temperature change from other anthropogenic forcing, (as in GeoMIP; Kravitz et al., 2011, described

in more detail below). It is therefore useful to develop emulators that can use existing simulations in order to predict climate consequences both for different future trajectories of greenhouse gas forcing and for different possible choices regarding the level of geoengineering.

The simplest emulator approach is pattern scaling (Santer et al., 1990; Mitchell, 2003; Tebaldi and Arblaster, 2014), where a predictive dynamic model is used only for the time-evolution of the global mean temperature (either from energy balance approaches or estimated directly from GCM simulations), and the temperature at every spatial location is assumed to vary with the same time evolution as the global mean – that is, that the pattern of temperature change is not itself a function of time.

Other variables, such as precipitation changes, are also assumed to depend only on the global mean temperature and on radiative forcing (Andrews et al., 2010); the only "memory" in the emulator in this case remains embedded in the dynamics of the global mean temperature response. Extending this, Cao et al. (2015) assume precipitation depends on global-mean-temperature and not just instantaneous $CO_2$ concentrations but also solar reduction, allowing for a different "fast" response to these

different forcings, but again maintaining global mean temperature as the sole dynamic predictor. Additional spatial patterns can also be included to capture other forcing agents including aerosols (Schlesinger et al., 2000; Frieler et al., 2012).

Of course, not all of the climate system responds to forcing with the same time-constants. Pattern scaling can be improved upon by introducing additional dynamic variables, such as land-sea tem-

perature contrast (Joshi et al., 2013), multiple empirical orthogonal functions (EOFs) of temperature (Holden and Edwards, 2010; Herger et al., 2015), or by including many more spatial degrees of freedom to better predict regional effects (Castruccio et al., 2014). The use of only one or a few dynamic variables (or predictors) is ultimately constrained by the difficulty in estimating the dynamic response of additional variables in the presence of climate variability due to low signal-to-noise ratio.

The primary assumption typically made in developing a climate emulator for predicting climate response is that the response is sufficiently linear and time-invariant (LTI). (We are explicit about our usage of the terms *linear* and *non-linear* in Section 2 below.) Success with emulators illustrates that linearity can be a reasonable approximation, although the accuracy of this assumption will depend on the variable and the level of applied forcing (e.g., Tebaldi and Arblaster, 2014). The response of

any LTI system to any time-varying forcing can be described by a convolution between the impulse response function that describes the system dynamics and the exogenous forcing; see equation (1) in Section 2 below, and also Åström and Murray (2008, Sec. 5.3) or Ragone et al. (2015, eq. 2). "Training" a linear emulator amounts to estimating the impulse response from one or more simulations. Nonlinear approaches to emulators are used in other aspects of climate modeling, such as

model tuning and parametric uncertainty analysis (Neelin et al., 2010), but such investigations are beyond the scope of this manuscript.

We start from the same LTI assumption here as in the references above, but extended to include solar geoengineering. The spatial patterns of the responses to solar and greenhouse gas forcing will

not be the same, leading to regional differences in outcomes (Ricke et al., 2010; Kravitz et al., 2014,
2015), nor are the precipitation responses the same (Bala et al., 2010; Andrews et al., 2010), nor
necessarily the time-evolution of the responses (Cao et al., 2015). All of these factors are important
to capture if the emulator is to be useful in understanding climate effects of strategies that include
solar geoengineering. We therefore only make an LTI assumption, and do not start with any addi-
tional *a priori* assumptions on the form of the dynamics. We thus consider independent predictors
for each variable. For estimating the spatial temperature and precipitation response, we employ an
EOF-based approach (as in Herger et al., 2015) with a common set of EOFs constructed from both
$CO_2$-forced and geoengineering simulations. In addition to temperature and precipitation, we also
consider Northern hemisphere sea-ice extent; the minimum extent over the year provides an example
where linearity is not a good assumption.

We use simulations from the Geoengineering Model Intercomparison Study (GeoMIP, Kravitz
et al., 2011) where solar reduction is used as a proxy for any approach that reduces incoming short-
wave radiation. Linearity and time-invariance are the *only* assumptions we make in developing the
emulator. The emulator can therefore be uniquely specified based on a single simulation for each
model. The assumption of linearity can then be evaluated by comparing predictions with a second
simulation for a different forcing trajectory; deviations between these result from nonlinearity, and
conversely, agreement validates linearity being a reasonable approximation. Section 2 describes the
methodology and simulations used, and the resulting emulator and validation are given in Section 3.

## 2 Approach

The expectation that an emulator calibrated to match the GCM response to one climate forcing path-
way can also do so for a different pathway is typically based on the assumption that the response to
forcing can be reasonably approximated as linear and time-invariant (LTI). Consider a system forced
by both time-dependent forcing $f(t)$ from changes in atmospheric greenhouse gas concentrations
and time-dependent forcing $g(t)$ from solar geoengineering. For any variable $z_i(t)$, define $z_i^f(t)$ as
the response to forcing $f(t)$ with $g(t) = 0$ and $z_i^g(t)$ as the response to forcing $g(t)$ with $f(t) = 0$,
where the response is defined in each case as the difference relative to the initial state, and neglect-
ing natural variability. The system is *linear* if for any scalars $\alpha$ and $\beta$, the response to the combined
forcing $\alpha f(t) + \beta g(t)$ is the same linear combination of the individual responses, $\alpha z_i^f(t) + \beta z_i^g(t)$.
Note that in general, even if the system is linear, the ratio of any two variables will vary with time
simply because different variables respond at different rates; that is, for any forcing scenario, there
is not in general some constant $\mu$ such that $z_i(t) = \mu z_j(t)$ for all time (a plot of $z_i(t)$ against $z_j(t)$
will not be a straight line if these variables respond with different time constants). The usage of the
word nonlinear to express this latter idea is distinct from the concept of the dynamic system itself

being linear or nonlinear. By a dynamic system we simply mean that $z(t)$ depends on past values of
the forcing $f(t)$ or $g(t)$ in addition to the current values.

The climate system as a whole is highly nonlinear. However, the response to a perturbation about
the current state may be close to linear; if the perturbation is sufficiently small then linearity will be
a good approximation.

### 2.1 Impulse Response

For an LTI system forced by both time-dependent $f(t)$ and $g(t)$, the response of any variable $z_i(t)$
can be expressed in terms of a convolution between the input time-series and the system impulse
response functions as

$$z_i(t) = \int_0^t h_i^f(\tau)f(t-\tau)d\tau + \int_0^t h_i^g(\tau)g(t-\tau)d\tau + n_i(t) \tag{1}$$

where $h_i^f(t)$ is the impulse response due to greenhouse gas forcing and $h_i^g(t)$ the impulse response
due to solar reductions; these will not in general be identical, nor in general the same for any choice
of output variable $z_i$. The variable $n_i(t)$ is included to capture the effects of climate variability.
Because the emulator is designed to capture the forced response, the actual character of $n_i(t)$ is
unimportant in defining the emulator. The system is time-invariant if $h_i^f(\tau)$ and $h_i^g(\tau)$ in equation (1)
do not depend explicitly on the current time $t$; some possible exceptions are noted in Section 4.
Note that the response of a linear system can be completely characterized by the impulse response;
knowing the impulse response is thus sufficient to predict the response to any forcing trajectory. The
same formalism would also apply for predicting the response to seasonally-dependent forcing, but
of course additional training simulations would be required.

If the climate system were indeed LTI, then equation (1) would hold for any variable (temperature,
precipitation, etc.), at global or regional scale, and whether annual-mean or at a shorter time-scale,
although the degree to which the forced response can be estimated in the presence of natural vari-
ability will vary with spatial and temporal scale, as will the influence of nonlinearities. We consider
variables evaluated once per year (e.g., annual-mean, or September sea ice extent), and equation (1)
can be cast in discrete-time to predict the response in year $k$ as

$$z_i(k) = \sum_{j=0}^k h_i^f(j)f(k-j) + \sum_{j=0}^k h_i^g(j)g(k-j) + n_i(k) \tag{2}$$

To estimate the impulse response for $CO_2$ forcing, we use the difference between the abrupt
$4\times CO_2$ simulation and pre-industrial simulation for each of the models participating in GeoMIP.
To estimate the impulse response for solar reduction, we use the G1 simulation from GeoMIP, in
which the $CO_2$ concentration was quadrupled and insolation decreased to approximately maintain
radiative balance and hence global mean temperature (see Figure 1). The difference between G1
and the $4\times CO_2$ simulations thus gives the response to an abrupt change in solar forcing, assuming

linearity. Note that each model separately chose the level of solar reduction $g_{4\times}$ required to balance
the forcing from increased atmospheric $CO_2$, so that the percent solar reduction in G1 varies from
model to model based on the efficacy of solar forcing in that model; see Table S1. Define

$$f(t) = \log_2 \left( \frac{CO_2(t)}{CO_{2,\text{ref}}} \right) \div 2 \tag{3}$$

$$g(t) = - \left( \frac{\text{Solar}(t) - \text{Solar}_{\text{ref}}}{\text{Solar}_{\text{ref}}} \right) \div g_{4\times} \tag{4}$$

where $CO_2(t)$ is the time-varying atmospheric $CO_2$ concentration and $\text{Solar}(t)$ is the solar irradi-
ance. The $4\times CO_2$ experiment then corresponds to forcing $f(t) = 1$, $t \geq 0$ and $f(t) = 0$, $t < 0$ with
$g(t) = 0$, while the GeoMIP G1 simulation uses the same $f(t)$ but with $g(t) = 1$, $t \geq 0$.

Substituting into Equation (2) then for any variable $z_i(k)$, the difference $z_i^{4\times}(k)$ between its value
in $4\times CO_2$ and preindustrial is given by

$$z_i^{4\times}(k) = \sum_{j=0}^{k} h_i^f(j) + n_i(k) \tag{5}$$

and the difference $z_i^{G1}(t)$ between its value in G1 relative to $4\times CO_2$ is

$$z_i^{G1}(k) = \sum_{j=0}^{k} h_i^g(j) + n_i(k) \tag{6}$$

from which we can estimate

$$\hat{h}_i^f(k) = z_i^{4\times}(k) - z_i^{4\times}(k-1) \qquad \text{and} \qquad \hat{h}_i^g(k) = z_i^{G1}(k) - z_i^{G1}(k-1) \tag{7}$$

The impulse responses $h_i^{f,g}(k)$ could be estimated from the time-series of any forced simulation, but
take particularly simple form from these step response simulations. (A linearly increasing forcing
scenario such as a 1% per year increase in $CO_2$ also leads to a simple form, with the continuous-time
impulse response proportional to the second derivative of the $1\%CO_2$ response.)

These impulse response estimates are "noisy" due to natural variability. Various approaches could
be used to reduce the influence of natural variability, such as

  1. Using multiple ensemble members or multiple forcing scenarios (as in Castruccio et al., 2014,
    for example),

  2. Only considering spatial averages by computing the global mean as in pattern scaling, pro-
jecting onto EOFs as in Herger et al. (2015), or averaging over specific spatial regions as in
    Castruccio et al. (2014),

  3. Applying temporal filtering to smooth high-frequency noise in $\hat{h}$ or fitting $h(t)$ to some esti-
    mated functional form such as semi-infinite diffusion for global mean temperature (Caldeira
    and Myhrvold, 2013) or a multiple-exponential (Castruccio et al., 2014) or

4.  Finding some less-noisy predictive variable, such as global mean temperature, to use as the predictor of other, noisier variables (effectively what is done in predicting the regional precipitation or temperature response in any pattern scaling analysis).

Choosing simulations with high forcing levels to train the emulator ($4\times CO_2$ and GeoMIP G1) increases the "signal" of the forced-response relative to the "noise" of climate variability. This choice
allows us to make useful predictions at lower forcing levels without the need for introducing additional assumptions on the functional form of the dynamics, such as that every field simply scales with global mean temperature. The penalty for this choice is that the high forcing will exacerbate any nonlinear effects; this choice precludes, for example, useful predictions of the Northern hemisphere annual-minimum sea ice extent (see Section 3 below), which would require that a lower-forcing
simulation be used to train the emulator.

A frequency-domain perspective is useful to understand how the "noise" due to climate variability affects the emulator predictions. The Laplace transform of Equation (1) transforms the convolution into multiplication:

$$\mathcal{L}(z_i) = \mathcal{L}(h_i^f)\mathcal{L}(f) + \mathcal{L}(h_i^g)\mathcal{L}(g) + \mathcal{L}(n_i) \tag{8}$$

$$= H_i^f(s)F(s) + H_i^g(s)G(s) + N(s) \tag{9}$$

where the Laplace transform of the impulse response, $H_i(s) = \mathcal{L}(h_i)$, is the *transfer function* between that input and that output; capital letters will denote the Laplace transform of $h(t)$, $f(t)$ and $g(t)$. (The discrete-time formalism in equation (2) could similarly be analyzed with a $Z$-transform; we use the continuous-time formulation here as readers are more likely to be familiar with it.) The
impulse response could thus equivalently be estimated by first taking the Laplace transform of the input and output, computing the ratio, and computing the inverse transform. Consider for example the response to increased $CO_2$ (the estimation for solar reduction is analogous), where the emulator is trained on the input $f_e(t)$ and used to predict the response to a different forcing time-series $f_p(t)$, with Laplace transforms $F_e(s)$ and $F_p(s)$. The transfer function estimate used by the emulator is

$$\hat{H}_i^g(s) = H_i^f(s) + \frac{N(s)}{F_e(s)} \tag{10}$$

and hence in the frequency domain the response predicted by the emulator for input forcing $F_p(s)$ is

$$\hat{Z}_i = Z_i(s) + N(s)\frac{F_p(s)}{F_e(s)} \tag{11}$$

That is, climate variability in the simulation used to train the emulator leads to an error in the pre-
diction that depends on the ratio of frequency content in the forcing signals between training and prediction simulations. Because a "step" change in the input such as in the abrupt $4\times CO_2$ simulation has more signal energy at low frequencies than high (Laplace transform proportional to $1/s$), it leads to a better estimate of the output response at low frequencies than at high frequencies; the

high-frequency estimation errors due to natural variability manifest as "noise" on the estimated impulse response (see Figure 2 for an example). However, the smoothly varying radiative forcing input due to a 1% per year increase in $CO_2$ has even less energy at high temporal frequencies than the step input (Laplace transform proportional to $1/s^2$). Thus training an emulator on a "step" input simulation and then using it to predict the results from a smoothly-varying forcing trajectory will result in relatively noise-free emulator predictions, despite the apparent high-frequency "noise" in the impulse response. Note that the GeoMIP G2 simulation (described at the beginning of the next section) has an abrupt change in the solar forcing at year 50 (see Figure 1), and the emulated responses to this "step" change in forcing are, as expected, noisier than those due to the smooth forcing changes over the first 50 years of G2.

## 2.2 Spatial analysis

For predicting the spatial pattern of the forced response, we estimate impulse responses not for every individual grid cell in each GCM, but only for the spatial response projected onto the first few empirical orthogonal functions (EOFs). For each model, EOFs are constructed from the area-weighted spatial temperature and (separately) the precipitation response. For each variable, and for each model, a single set of EOFs is constructed using output from both the $4\times CO_2$ and G1 simulations, leading to a description of the form

$$T(x,y,t) = \sum_{i=1}^{m} \Phi_i(x,y)\psi_i(t) \tag{12}$$

where $\Phi_i$ are the spatial basis functions (EOFs) and $\psi_i$ the corresponding principal components (projection of $T(x,y,t)$ onto each $\Phi_i$ for any particular forcing scenario); the basis set $\Phi_i$ are thus unchanged across the different forcing mechanisms and temporal trajectories. Truncating the set of EOFs provides a maximally efficient basis for describing the spatial pattern of the response, capturing any pattern strongly excited by either one or both forcing mechanisms. In general, only the first few principal components are distinguishable from climate variability and have any predictive capability (Figure S4) and we retain $m = 4$ throughout. The first pattern, corresponding to the highest variance in the simulations, is similar to the long-term pattern of global warming; choosing $m = 1$ would thus be analogous to pattern scaling. Including additional EOFs captures both the differences in how the climate responds to solar versus $CO_2$ forcing, as well as differences between the short- and long-term pattern of response for either forcing (i.e., that not everything responds at the same rate). Temperature EOFs for one model are shown in Supplementary Material Figure S1, where the second EOF captures the equator-to-pole differential warming that is a robust signature of compensating a $CO_2$-induced global mean temperature rise with a solar reduction, while EOFs 3 and 4 capture Northern hemisphere and global patterns of land temperature, which change more rapidly than ocean temperatures in response to forcing.

The impulse responses can then be separately estimated for each principal component as before from the $4\times CO_2$ and G1 simulations, and the time series of $\psi_i$ for any other forcing scenario estimated. Equation 12 is then used to construct the estimate of the spatial response.

## 3 Results and validation

The impulse responses $h_i^f(t)$ and $h_i^g(t)$ are estimated for a number of different variables from the abrupt $4\times CO_2$ and G1 simulations as described above. The impulse-response based emulator for $CO_2$ forcing without any solar reduction can be validated by comparing the predictions with the simulations for a 1% per year increase in $CO_2$ (1%$CO_2$). To validate the emulation of solar reduction, we use the GeoMIP G2 scenario, in which $CO_2$ levels increase at 1% per year, and for the first 50 years, the solar reduction is gradually increased to balance this forcing. This uses the same ratio of $g(t)$ to $f(t)$ as in G1 for each model. After 50 years, the solar reduction is returned to zero so that only the radiative forcing from the $CO_2$ remains (see Kravitz et al. (2011) and Figure 1 for a schematic of the forcing in the G1 and G2 simulations). Several of the climate models that conducted experiments G1 and G2 exhibit significant drift in the absence of net radiative forcing, due to the initialization state not being in equilibrium. These models are not considered further, leading to a total of 9 models considered here (Table S1).

The impulse response functions for predicting the global mean temperature and precipitation responses to either $CO_2$ or solar forcing are shown in Figure 2, averaged over all of these climate models (see Supplementary Material for tabulation of these and other impulse responses for each model). As expected these are "noisy" estimates due to natural variability. Note that while the temperature response characteristics are similar (aside from the sign) for increased $CO_2$ and reduced insolation, the precipitation response differs. The impulse response of precipitation clearly highlights that while $CO_2$ and solar reduction have a similar "slow" response (changes in precipitation that result from changes in temperature), they have quite different "fast" responses (rapid atmospheric adjustments in the climate system before temperature has time to adjust). The fast response is related to different amounts of radiative forcing absorbed by the atmosphere that affect stability and convection (e.g., Andrews et al., 2010). For $CO_2$-forcing this leads to an initial precipitation response of the opposite sign to the long-term slow response; while solar reductions might largely compensate for the slow response there will be residual differences due to the differential fast response. Comparing impulse response functions between models may also be useful to identify differences in dynamics (Figure S1).

Figure 3 validates the ability of the impulse response formulation in equation (1) tuned from the $4\times CO_2$ and G1 simulations to correctly predict the global mean temperature response from the 1%$CO_2$ and G2 simulations. Linearity has previously been argued as a reasonable assumption for temperature and precipitation responses (Kravitz et al., 2014, and references therein). Since that is

the only assumption made in constructing the emulator, the error in estimating the forced response arises only from natural variability and from nonlinearity. The difference between GCM-simulated and emulator-predicted trajectories is similar to the standard deviation of natural variability in many models; see Table 1. Cases where the predicted and simulated responses agree to within the limit imposed by natural variability indicates that nonlinear effects are small relative to variability, and hence this analysis also illustrates the utility of a linearity assumption at these forcing levels.

Figure 4 shows the corresponding plots for global mean precipitation. The deviation between emulated and simulated responses are higher for some models here than for temperature, though the estimation errors are close to the limit due to natural variability for many models. Note that since G2 suppresses global mean temperature changes, it largely suppresses the slow (temperature-dependent) precipitation response (there will still be some effect from regional temperature changes). This suggests that in models such as GISS-E2-R, HadCM3, or MIROC-ESM where the G2 emulation is notably better than the emulation of the 1% $CO_2$ simulation, larger nonlinearities in the precipitation response arise in the slow rather than fast response to precipitation.

Similar results are shown in the supplementary material (Figures S2–S3) for the temperature or precipitation difference between land and ocean; the only notable case where the error from nonlinearity exceeds natural variability is in the GISS-E2-R prediction of land-sea precipitation differences in the 1% $CO_2$ simulation. While it is not our purpose to evaluate mechanisms of nonlinearity in the climate models, this type of analysis may be useful input into such research.

Northern hemisphere sea ice extent is an example of a variable that is both highly relevant for assessing possible future scenarios, yet one in which a nonlinear response to forcing might be expected. The 4×$CO_2$ forcing is large enough that September sea ice is nearly lost in all models, and thus an emulator trained off of this simulation will do a relatively poor job at predicting the reduction in annual-minimum sea ice extent from smaller forcing; see Figure 5. However, despite the obvious nonlinearity in the annual-minimum extent, the annual-mean sea ice extent does behave sufficiently linearly in most models, even at this large a forcing level, so that the 4×$CO_2$ simulation can be used to train a useful emulator. This is illustrated in Figure 6.

Finally, Figure 7 illustrates the ability to capture the spatial response. One of the concerns raised regarding the use of solar geoengineering is that the response from solar reduction does not perfectly compensate that from increased $CO_2$, resulting in some regional differences in temperature and precipitation responses (Ricke et al., 2010; Kravitz et al., 2014). It is therefore valuable to assess whether the emulator can capture some of the regional variation in the response between $CO_2$ and solar forcing. As described earlier, the regional response is predicted using EOF analysis and estimating the forced-response for the first few principal components. Figure 7 plots the model-mean temperature and precipitation responses averaged over years 41-50 of the G2 simulation for both the simulation and the emulator prediction. The G2 simulation, like G1, results in overcooling of the tropics and undercooling of the poles. The emulator slightly underpredicts the residual Arctic

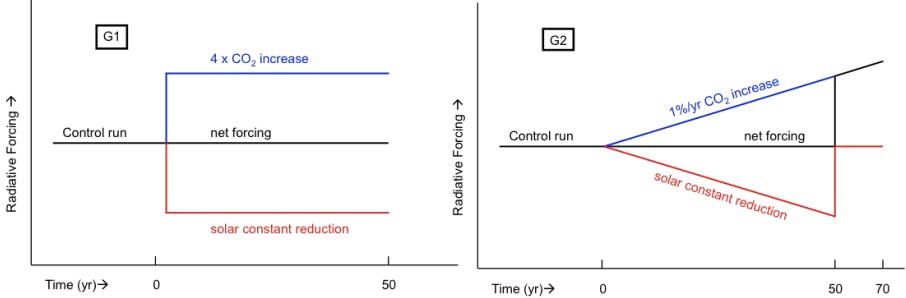

**Figure 1.** Schematic of GeoMIP G1 and G2 simulations, from Kravitz et al. (2011).

warming in G2, likely due to the nonlinearity associated with sea ice albedo feedback at the $4\times CO_2$ forcing used in training the emulator. The area-weighted spatial root mean square (rms) of the difference between emulated and simulated responses is also shown in Table 1, normalized at each grid cell by the standard deviation of interannual climate variability. Where the rms value is close to unity implies that the errors introduced by assuming linearity are not limiting the emulator predictions; the

Arctic nonlinearity contributes to the larger rms errors in temperature prediction for many models.

     This raises an interesting observation. If it is purely the forced-response that is of interest, then a single GCM simulation of a low-forcing scenario such as G2 leads to uncertainty in the estimate due to natural variability. While the most accurate estimate would be obtained by averaging over a sufficiently large ensemble, this may not be achievable for computational reasons. The emulator

provides a computationally-efficient alternative. Because the emulated response is based on simulations with roughly three times higher radiative forcing, and because the process of its construction suppresses high-frequency natural variability (equation 11), the estimate of the forced-response that it provides has less uncertainty due to natural variability, at the cost of increased errors from nonlinearity. It is thus possible that, given only sufficient computation to conduct a single simulation, the

emulated response based off of G1 could be a more accurate representation of the forced-response to G2 than that obtained from the actual G2 simulation. This is trivially true if indeed the response was perfectly linear; in general there is a trade-off between errors due to nonlinear effects and the uncertainty introduced by variability.

## 4    Discussion

Climate emulators provide a powerful tool for assessing any proposed future pathway of mitigation choices (including carbon dioxide removal) and different levels of geoengineering. For example, solar geoengineering could be used only to limit peak warming as part of an "overshoot" scenario in which atmospheric $CO_2$ concentrations peak and subsequently decline as net-negative carbon emissions reduce concentrations (Long and Shepherd, 2014; Tilmes et al., 2016). A limited, temporary

deployment has also been described as a way to reduce the rate of warming (Keith and MacMartin,

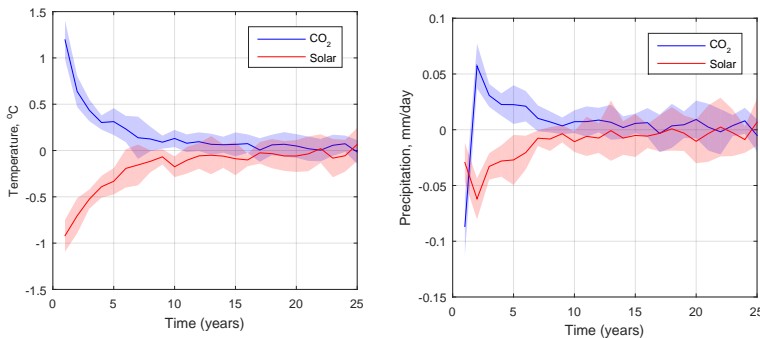

**Figure 2.** Estimated impulse response for $CO_2$ and solar forcing, for global mean temperature and precipitation, averaged over all 9 models (Table S1); the inter-model standard deviation is shown by the shaded bands. While these impulse response functions are "noisy", predictions made using them are less so, particularly for forcing levels much smaller than those used in estimating these functions. Note for precipitation the robust "fast" response to increased $CO_2$ has the opposite sign as the "slow" response. Temperature and precipitation units are given as the response for a quadrupling of $CO_2$. (See Supplementary Material including Figure S1 for individual model impulse respones functions.)

| Model | Global-mean Temperature | | Global-mean Precipitation | | Annual mean NH sea ice | | Spatial rms Temperature | | Spatial rms Precipitation | |
|---|---|---|---|---|---|---|---|---|---|---|
| | 1%$CO_2$ | G2 | 1%$CO_2$ | G2 | 1%$CO_2$ | G2 | 1%$CO_2$ | G2 | 1%$CO_2$ | G2 |
| CanESM2 | 1.0 | 0.5 | 1.3 | 0.5 | 1.8 | 1.3 | 1.4 | 1.2 | 1.0 | 0.7 |
| CESM-CAM5.1-FV | 1.4 | 1.0 | 1.0 | 0.7 | - | - | 1.8 | 1.2 | 1.5 | 1.2 |
| GISS-E2-R | 1.1 | 1.2 | 3.0 | 2.0 | 1.6 | 1.4 | 2.2 | 1.8 | 2.2 | 1.3 |
| HadCM3 | 1.3 | 1.3 | 3.2 | 1.3 | - | - | 2.2 | 1.9 | 1.3 | 1.2 |
| HadGEM2-ES | 1.4 | 1.7 | 2.7 | 1.3 | 1.0 | 1.7 | 2.4 | 1.7 | 1.0 | 0.8 |
| IPSL-CM5A-LR | 1.2 | 1.9 | 1.2 | 1.7 | - | - | 2.6 | 2.0 | 2.0 | 2.1 |
| MIROC-ESM | 1.8 | 0.7 | 1.3 | 1.1 | 2.2 | 1.3 | 4.0 | 1.5 | 2.4 | 1.4 |
| MPI-ESM-LR | 1.7 | 0.8 | 1.3 | 0.9 | - | - | 2.6 | 1.2 | 1.5 | 1.1 |
| CSIRO-Mk3L-1.2 | 1.4 | 1.7 | 1.2 | 0.7 | - | - | 4.1 | 1.3 | 3.7 | 0.8 |

**Table 1.** Root-mean-square (rms) deviation between simulation and emulator prediction. For first three (scalar) variables, temporal rms is computed over years 31–50, normalized by the standard deviation of interannual natural variability. For spatial response, the area-weighted rms is computed after normalizing by variability at each grid cell (that is, the spatial rms of the deviation as measured in standard deviations of natural variability).

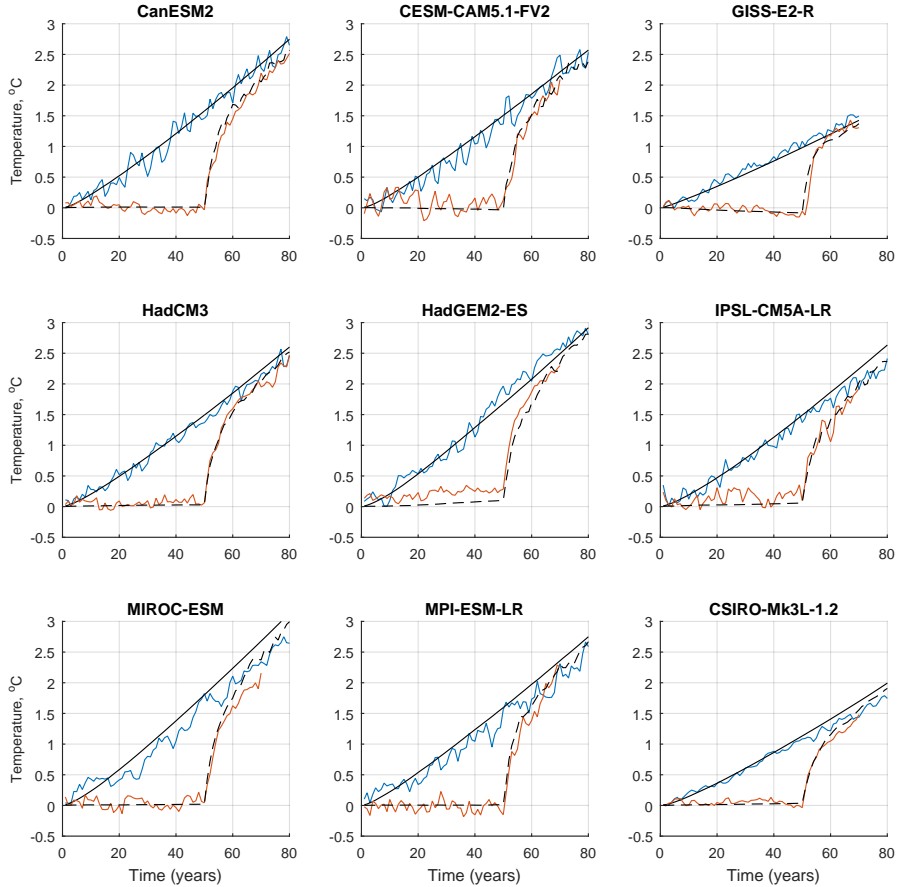

**Figure 3.** Simulated and predicted global mean temperature, both for a 1% per year increase in $CO_2$ (blue curves) and for GeoMIP experiment G2 (red), for each of the climate models considered here. The predicted response using the emulator is given by black lines, solid for the 1% $CO_2$ case and dashed for G2.

2015; MacMartin et al., 2014). These types of limited-deployment scenarios are motivated in part by recognizing that solar geoengineering sufficient to reduce global mean temperature to preindustrial levels could lead to significant regional disparities and other risks, while a deployment that only partially reduces global mean temperature might decrease some metrics of climate change everywhere
(Kravitz et al., 2014).

By training emulators on a standard set of simulations, such as GeoMIP, that have been conducted by multiple modeling centers, any proposed scenario such as these can be readily evaluated with multiple models. This yields a computationally-efficient method for providing insight into the robustness of conclusions. (Of course, any collection of models is an ensemble of opportunity, with
interpretation challenges as a statistical sample; see, e.g., Collins et al. (2013), Section 12.2, for a thorough discussion.) The emulator used here assumes that the climate system response can be sufficiently well approximated over the range of forcing levels of interest by the output of a linear

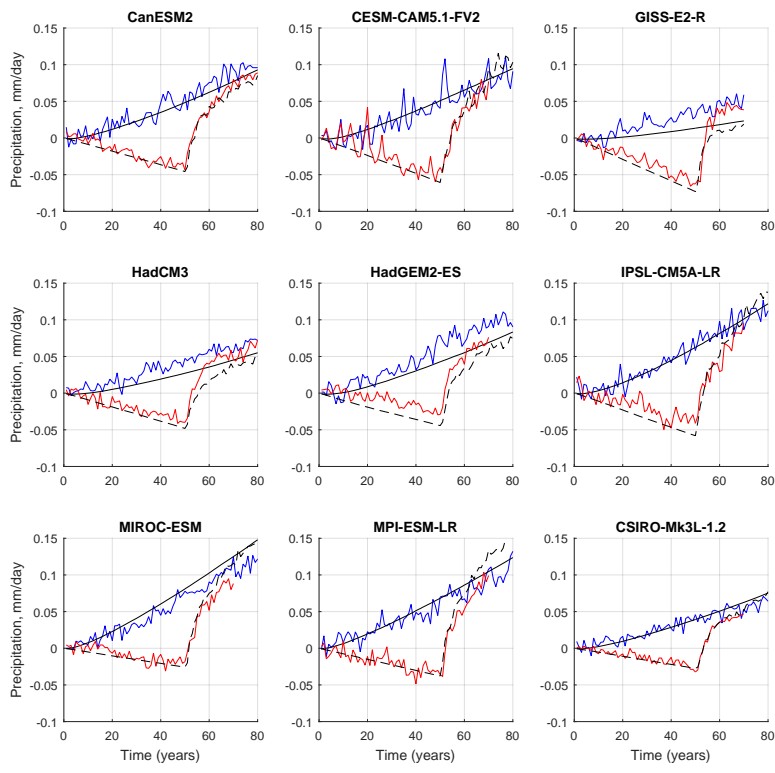

**Figure 4.** As in Figure 3 but for global mean precipitation. Simulated and emulated response are shown for 1% per year increase in $CO_2$ and GeoMIP experiment G2 for each of the climate models considered here.

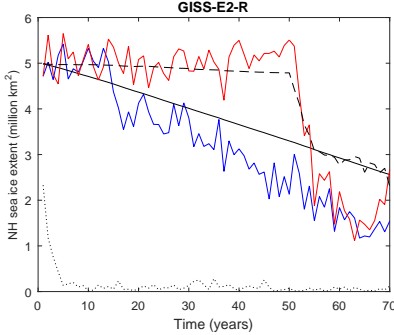

**Figure 5.** As in Figure 3 but for Northern Hemisphere annual-minimum sea ice extent. Simulated and emulated response are shown for 1% per year increase in $CO_2$ and GeoMIP experiment G2 for one model, GISS E2-R. The dotted line shows the response for the abrupt $4\times CO_2$ simulation. The relatively poorer emulator prediction for the 1% $CO_2$ case in particular illustrates that the linearity assumption does not hold for all relevant climate variables.

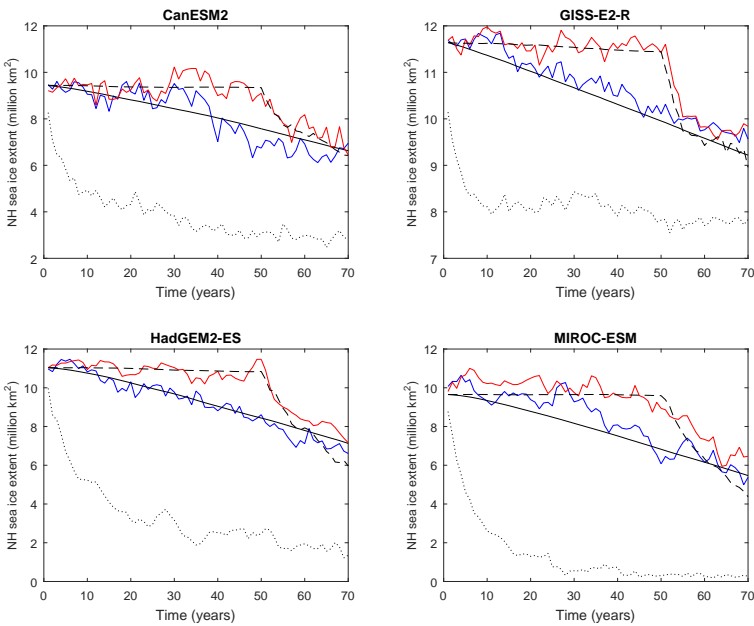

**Figure 6.** As in Figure 3 but for Northern Hemisphere annual-mean sea ice extent. Simulated and emulated response are shown for 1% per year increase in $CO_2$ and GeoMIP experiment G2 for several of the climate models considered here; the dotted line shows the response for the abrupt $4 \times CO_2$ simulation.

system. For many variables, the analysis here indicates that this is a sufficiently good assumption, with the difference between simulated and emulated responses similar to the standard deviation of natural variability. There are many more variables that may be of interest; similar analysis as here could be used to assess whether a linear assumption is or is not sufficient for projecting the response of any variable beyond those considered here. The GeoMIP simulations are also of limited duration, and nonlinearities may arise at longer time-scales due to changes in ocean dynamics, for example (Bouttes et al., 2015).

Finally, note that the results herein were obtained using simulations that reduce the solar constant as a proxy for any solar geoengineering approach. While this is clearly a useful first step, the climate effects from any specific technology, such as stratospheric aerosol injection (SAI) or marine cloud brightening (MCB) will differ (e.g., Ferraro et al., 2015) both due to the different mechanism of radiative forcing, and the different spatial pattern of radiative forcing (the latter being at least partially a design choice; Kravitz et al., 2016). Further, while linearity appears to be a reasonable assumption in these climate models for predicting the response of many climate variables to an imposed solar reduction, it may be a poorer approximation for SAI, for example. Nonlinearities will occur in aerosol size distribution (Heckendorn et al., 2009; Niemeier and Timmreck, 2015), as well as due to changes in the stratospheric circulation that result from the aerosols (Aquila et al., 2014);

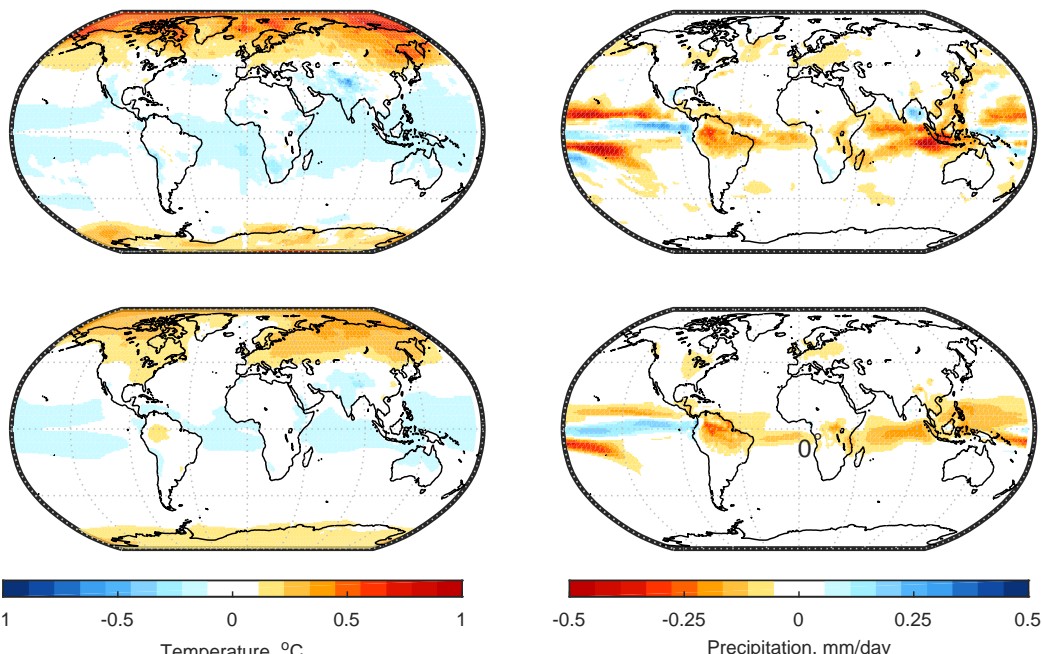

**Figure 7.** Temperature (left) and precipitation (right) averaged over years 41-50 of G2 simulation and averaged over all 9 models. The upper row shows the simulated results; the lower row shows the prediction based on a spatial emulator developed using 4 EOFs for each model. As noted elsewhere, the robust response to increasing $CO_2$ and reducing insolation to maintain zero global mean temperature difference is a net reduction (overcompensation) of global mean precipitation (Bala et al., 2010), and an overcooling of the tropics and an undercooling of the poles (Kravitz et al., 2013). The latter is an artifact of a latitudinally-uniform reduction in sunlight, and could be better managed by increasing the forcing at high latitudes relative to low (Kravitz et al., 2016).

time-invariance might also not hold if, for example, time-varying stratospheric chlorine concentrations (which affects the aerosol impact on ozone) are considered part of the "system" rather than a forcing. It is unclear how significantly these will affect the ability to develop emulators for this technology.

*Author contributions.* DGM and BK designed the study, conducted the analysis, and wrote the paper.

*Acknowledgements.* We thank all participants of the Geoengineering Model Intercomparison Project and their model development teams, CLIVAR/WCRP Working Group on Coupled Modeling for endorsing GeoMIP and the scientists managing the Earth System Grid data nodes who have assisted with making GeoMIP output available. The Pacific Northwest National Laboratory is operated for the U.S. Department of Energy by Battelle Memorial Institute under contract DE-AC05-76RL01830. This work was partially supported by Cornell
University's David R. Atkinson Center for a Sustainable Future (ACSF).

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
