# Peer review of "Dynamic climate emulators for solar geoengineering"

_Atmospheric Chemistry and Physics, 2016_

## Referee Comment (RC1) · Anonymous Referee #1 · 1 Jul 2016

The authors propose a nonparametric emulator aimed at reproducing geoengineering scenarios. Using dynamical linear models, they propose a formulation of the emulator as a convolution of the forcing and the impulse responses, and test this approach for two geoMIP scenarios for some variables of interests.

The manuscript is overall well written and presents an interesting problem, but I believe that in its present form is not suitable for publication and, in order to be reconsidered, needs to be considerably improved in many parts. The proposed method would have considerable limitations if it is to be expanded beyond the narrow context of this work, e.g. annual averages for two model runs. Further, the validation setting is extremely limited, not based on any metric, and completely ignores the emulation uncertainty.

[Figure]

**General comments**

- The validation setting is extremely limited: the proposed approach is fit for the G1 scenario, and then used to extrapolate G2. Also, for the G1 scenario the emulator is likely to work well, since it consists of impulse functions. A considerable amount of work is needed to perform more tests under different forcing scenarios. While the geoMIP is limited in size, the CMIP5 or other large multi-model ensembles could be used to validate the forcing part of this emulator.

- The present version puts very little emphasis of the uncertainty of the scenario estimation. The validation essentially consists in eyeballing many plots of the emulator against the original computer model, with no attempt to quantify the fit or, most importantly, to asses how the internal variability of the model is reproduced by the emulator. The definition itself of 'climate variability' $n_i(t)$ is unclear. Are the authors assuming a white noise? Also, I would assume that this noise is independent for different variables, but it should be clearly stated.

- This approach will have significant limitations at finer temporal scales. The authors briefly discuss this when they mention how we can impose $h = h(\tau, m)$. This solution is not straightforward, as a nonparametric estimation of 12 different impulse responses will require more scenarios (surely more than two) to have reliable estimates. The authors somewhat acknowledge it when they state that additional simulations would be required, but in an off-the-shelf ensemble such as geoMIP, where no more scenarios are readily available, this is a strong limit of this approach. This will be become even more evident for finer temporal scales, e.g. weekly or daily data.

- The results and the discussion do not mention model differences, and most importantly what do they mean. Does the emulator estimate different impulse responses for different models? I would expect so, and I would expect these differences to convey information on how the models differ. For example, HadCM3 and HadGEM2-ES will likely display similar responses as both models are released from the Hadley Centre.

- The part on grid-scale emulation must be extended. Firstly, the methodology is unclear: a clear explanation of how were the EOFs selected must be presented, either in the main text or in the supplement. Secondly, as before, a more formal assessment of the pattern similarity is needed, as eyeballing figure 5 is not enough to convince that the emulator is performing well.

**Specific comments**

- Title: what the authors present is not a multi-model emulator, in the sense that it independently fits each model and does not assume interdependencies.

- pag. 1 l.16-17. The claim that the 'emulator prediction may be a more accurate estimate [...] of the models' response than an actual simulation' is very questionable. The emulator is not meant to replace a climate model, it's just a faster approximation that is used to explore the input space in a computationally efficient manner. While emulators are arguably a useful tool for calibration and, as in this case, scenario extrapolation, they cannot replace the physics of the climate model and they are useful only as long as the training set from the climate model is meaningful.

- pag 1. l.19-20. Actually, emulators are much more popular in model calibration and local sensitivity analysis of physical parameters then in projections of anthropogenic forcings. Only very recently this methodology have been extended to deal with forcings. This introductory part must be rewritten with a more extensive literature review on traditional emulators.

- pag. 4, eq (1) and onwards. It is somewhat inappropriate to represent the emulator as a convolution given that the authors are effectively using just annual averages. A reformulation in terms of discrete sums is necessary.

- pag. 4, line 101. $h(\tau)$ was never defined.

- pag. 6, line 161. Poor choice of pedix in $f_t(t)$, please reformulate.

- Figures. What is the unit measure of precipitation? Also, are the all figures expressed as anomaly with respect to a reference value? If so, what is it?

---

## Author Comment (AC1) · 21 Jul 2016

**Response to Anonymous Referee #1 (our comments in blue)**

The authors propose a nonparametric emulator aimed at reproducing geoengineering scenarios. Using dynamical linear models, they propose a formulation of the emulator as a convolution of the forcing and the impulse responses, and test this approach for two geoMIP scenarios for some variables of interests.

The manuscript is overall well written and presents an interesting problem, but I believe that in its present form is not suitable for publication and, in order to be reconsidered, needs to be considerably improved in many parts. The proposed method would have considerable limitations if it is to be expanded beyond the narrow context of this work, e.g. annual averages for two model runs. Further, the validation setting is extremely limited, not based on any metric, and completely ignores the emulation uncertainty.

> We agree that we could do a better job of quantifying emulator accuracy (i.e., evaluating based on specific metrics); we will address this in revision.

> The description in the paper of how to extend results to cover sub-annual time-scales was poorly worded and thus misleading. The case of evaluating sub-annual variables in response to annual-mean forcing is already covered by the method (indeed we considered one such variable but did not include a plot in the paper, we will do so in revision). The description in the paper regarding sub-annual scales was intended to refer to the case where the forcing varies at sub-annual time-scales. Conceptually this is a trivial extension, but would require additional training data; this is not a limitation of our method per se, but only a limitation that information cannot be invented from nothing. (You cannot infer the response to seasonally-varying forcing from a simulation where the forcing was constant, no matter what method you apply.)

> Also note that our formalism for capturing the linear forced response is intentionally provided in a more general fashion than most previous research. The key distinction relative to the existing literature aimed at the similar problem for non-geoengineering forcing is that we only assume linearity, and do not choose to make additional ad hoc (and apparently unnecessary) assumptions regarding the form of the dynamics; in doing so it helps clarify the additional assumptions made elsewhere. We will reword to better articulate the relationship with existing approaches.

> We disagree regarding the limitation of the validation setting, as described in more detail below. The perceived limitation regarding annual averages is not a limitation but simply poor wording on our part. Validation on one forcing scenario is sufficient to demonstrate that linearity is a sufficiently useful approximation for the variables that we consider here. Since that is the only assumption we make, additional validation scenarios would not add any further value.

**General comments**

• The validation setting is extremely limited: the proposed approach is fit for the G1 scenario, and then used to extrapolate G2. Also, for the G1 scenario the emulator is likely to work well, since it consists of impulse functions. A considerable amount of work is needed to perform more tests under different forcing scenarios. While the geoMIP is limited in size, the CMIP5 or other large multi-model ensembles could be used to validate the forcing part of this emulator.

> We demonstrated that a *linear* emulator trained on one scenario successfully follows the behaviour of another. Note that in contrast with other studies looking at climate emulators for greenhouse gas forcing, the *only* assumption we make is linearity (and time invariance). By demonstrating that linearity holds for forcing levels up to 4xCO2 and for levels of solar reduction sufficient to compensate this, it will clearly also hold for *any* other smaller-amplitude forcing scenario and hence no value would be added by validating it on additional forcing scenarios. We will reword the paper to clarify this.

• The present version puts very little emphasis of the uncertainty of the scenario estimation. The validation essentially consists in eyeballing many plots of the emulator against the original computer model, with no attempt to quantify the fit or, most importantly, to assess how the internal variability of the model is reproduced by the emulator. The definition itself of 'climate variability' $n_i(t)$ is unclear. Are the authors assuming a white noise? Also, I would assume that this noise is independent for different variables, but it should be clearly stated.

> We agree that we should do a better job of quantifying the fit, we will address this in revised manuscript.
>
> We also need to clarify in the revised text that the emulator is intended to only capture the forced response, and is not intended to reproduce internal variability. We will also clarify the text describing the effect of the spectral characteristics of the climate variability. Insofar as we are only interested in capturing the forced response, we do not need to make any assumptions about the nature of the climate variability as it enters only as "noise" in our ability to estimate the forced response; we will say this as well.

• This approach will have significant limitations at finer temporal scales. The authors briefly discuss this when they mention how we can impose h = h(_,m). This solution is not straightforward, as a nonparametric estimation of 12 different impulse responses will require more scenarios (surely more than two) to have reliable estimates. The authors somewhat acknowledge it when they state that additional simulations would be required, but in an off-the-shelf ensemble such as geoMIP, where no more scenarios are readily available, this is a strong limit of this approach. This will be become even more evident for finer temporal scales, e.g. weekly or daily data.

> As noted above, we apologize for badly worded text here that was potentially misleading. To clarify, if the *forcing* does not vary significantly over the course of the year, then emulating GCM response at finer temporal scales is not intrinsically more difficult for this or any other emulator, although the signal to noise ratio (SNR) will likely be poorer (a limitation that is intrinsic to the information contained in the training data, and has nothing to do with the method itself).
>
> If the intent is to capture the response to seasonally-varying forcing, then unless arbitrary assumptions are made regarding the seasonal dependence of the impulse response, one would need at least as many independent forcing scenarios as degrees of freedom of the seasonal response (i.e., 12 if one wants to distinguish how the response depends on monthly-varying forcing). This is not a limitation of the formulation we use (it would be a trivial extension), rather it is an intrinsic limitation on the knowledge of the response that holds for any such approach. Of course, for evaluating climate change in response to different pathways of greenhouse gas forcing and solar geoengineering, the forcing varies only slowly from year to year, so that the additional training data is not needed.
>
> We did a poor job of articulating the distinction between these two cases, and will correct this.
>
> We focused on information about annual-average behaviour because it is indeed useful, both for geoengineering and more general climate science applications although we could certainly include some sub-annual variables in revision (the only one we looked at in writing the paper was annual-minimum sea ice extent for which nonlinearity is significant and the emulator does not perform well; we will add this to the revised paper).

• The results and the discussion do not mention model differences, and most importantly what do they mean. Does the emulator estimate different impulse responses for different models? I would expect so, and I would expect these differences to convey information on how the models differ. For example, HadCM3 and HadGEM2-ES will likely display similar responses as both models are released from the Hadley Centre.

> Our intent was to develop an approach for emulating climate models, not for describing the differences between them, for which there is already an abundant literature on the general differences between climate models in terms of processes they include, differences in how they respond to climate change, and differences in how they respond to geoengineering. We will add some text to this effect and appropriate references, although given the vast breadth of this sort of literature, the number of citations we can include will naturally be limited..

• The part on grid-scale emulation must be extended. Firstly, the methodology is unclear: a clear explanation of how were the EOFs selected must be presented, either in the main text or in the supplement. Secondly, as before, a more formal assessment of the pattern similarity is needed, as eyeballing figure 5 is not enough to convince that the emulator is performing well.

We agree that this section was too terse. We will add both a more complete description, and a more formal pattern similarity assessment.

**Specific comments**
• Title: what the authors present is not a multi-model emulator, in the sense that it independently fits each model and does not assume interdependencies.

We agree that there are multiple ways of interpreting the phrase "multi-model". We meant it only in the sense that the end result is a set of emulators (i.e., in the same sense that CMIP or GeoMIP are "multi-model" ensembles.) We will clarify this in the revised version.

• pag. 1 l.16-17. The claim that the 'emulator prediction may be a more accurate estimate [...] of the models' response than an actual simulation' is very questionable. The emulator is not meant to replace a climate model, it's just a faster approximation that is used to explore the input space in a computationally efficient manner. While emulators are arguably a useful tool for calibration and, as in this case, scenario extrapolation, they cannot replace the physics of the climate model and they are useful only as long as the training set from the climate model is meaningful.

On the final point, we of course agree completely – the climate model is needed to generate training data for the emulator, and an emulator cannot completely replace climate models. We also agree that the emulator is only useful so long as the training set is meaningful. We will ensure that these points are clear in the revised version.

As to whether the emulator prediction is a more accurate estimate for some specific scenario, that depends both on the purpose and on the input/output response of the dynamic system being emulated. Our goal is to estimate the *forced* component of the response, isolated from natural variability, and in doing so we approximate the response as linear. A single GCM simulation of a particular scenario will not give a perfect estimate of the forced response, due to the presence of natural variability, while the emulated response, with an emulator trained from a simulation at higher forcing amplitude, will introduce some error due to nonlinearity, but *less* uncertainty due to natural variability. *Fundamentally one is simply trading off the uncertainty in the forced response that comes from superimposed natural variability from the uncertainty that comes from nonlinearity.* The best answer for the forced response would come from a sufficiently large ensemble of GCM simulations of the specific scenario, but given sufficient computation for one single simulation, then it is not a priori clear whether the best estimate of the forced response in a particular scenario is obtained by simulating that particular scenario… it may well be true that simulating at a higher forcing amplitude, to give a higher SNR, and then scaling the response, would indeed give a better estimate. (If the system were perfectly linear in its response to forcing, this is self-evident.) We will revise the manuscript to make this point more clearly.

• pag 1. l.19-20. Actually, emulators are much more popular in model calibration and local sensitivity analysis of physical parameters then in projections of anthropogenic forcings. Only very recently this methodology have been extended to deal with forcings. This introductory part must be rewritten with a more extensive literature review on traditional emulators.

We can add some references to acknowledge the prior history. However, it is only the recent extension to deal with forcings that is directly relevant to the case here.

• pag. 4, eq (1) and onwards. It is somewhat inappropriate to represent the emulator as a convolution given that the authors are effectively using just annual averages. A reformulation in terms of discrete sums is necessary.

Agreed that we should describe in terms of discrete sums; the continuous-time derivation was provided only because we felt that some readers might be more comfortable with it. We will fix this.

• pag. 4, line 101. h(_ ) was never defined.
• pag. 6, line 161. Poor choice of pedix in $f_t(t)$, please reformulate.

Thanks; we will clarify notation

• Figures. What is the unit measure of precipitation? Also, are the all figures expressed as anomaly with respect to a reference value? If so, what is it?

Oops.  Sorry, final version of figures were generated but didn't get included!  Not sure how that happened or passed final proof-reading.  Units are in mm/day, and are all in anomalies with respect to the preindustrial control values.

---

## Referee Comment (RC2) · Anonymous Referee #2 · 24 Sep 2016

The authors used model results from Geoengineering Model Intercomparison Project (GeoMIP) to test the linearity of the climate response to external forcings. The authors first constructed a climate emulator based on a convolution of impulse response function using results from GeoMIP G1 simulations involving abrupt changes in atmospheric $CO_2$ and solar irradiance. Then the authors used the climate emulator to predict climate consequences of the GeoMIP G2 simulations involving gradual change in atmospheric $CO_2$ and solar irradiance. For climate variables including temperature, precipitation, and annual mean Northern Hemisphere sea ice extent, the emulator does a good job in reproducing climate model simulated temporal evolution and spatial distribution. The use of impulse response function to emulate climate model results is not new. The novelty of this study is that it extends the application of impulse response function to the simulations involving both $CO_2$ and solar forcing. This extension advances our understanding of climate response to external forcing, and in particular, climate response to solar geoengineering. The ms is well written. I recommend publication after the following issues are addressed:

1. The GeoMIP simulations are limited to a period of 50 years. Over longer timescales (several centuries), response from deep ocean dynamics would become important. Many aspects of ocean dynamics response (e.g., thermohaline circulation) are nonlinear. So the question is: To what extend the linear emulator would be valid in reproducing long-term climate response involving feedbacks from deep ocean dynamics?

2. A large part of the residual response of the hydrological cycle over land to solar geoengineering is due to the direct effect of increasing atmospheric $CO_2$ on vegetation (stomatal, leaf area index, etc.), which cannot be offset by reduced solar forcing. Assumedly, this part of hydrological cycle response is nonlinear. This issue should be discussed.

3. The method used to emulate spatial pattern of temperature and precipitation is not clear. How EOFs were constructed, selected, and applied to generate the spatial pattern of climate change? These should be elaborated.

---

## Referee Comment (RC3) · Anonymous Referee #3 · 28 Sep 2016

The paper entitled "Multi-model dynamic climate emulator for solar geoengineering" by MacMartin & Kravitz presents a simple numerical emulator of the complex GeoMIP models which could be used to discuss Geoengineering scenarios. This paper is well written, fairly straightforward, and is interesting – I believe – for the community.

One could wonder, however, if ACP is the best journal for publication, as a lot of technical detail regarding the modeling (i.e. establishing the response functions) is given, whereas the more physical aspects remain (maybe too) brief. Maybe GMD would have been a better choice. But that is ultimately an editorial issue. And I don't think this point alone prevents publication in ACP, especially as the physics is well understood and already published elsewhere. It goes in favor, however, of improving the narrative so that the reader can grasp both the modeling approach and the modeled physical processes.

[Figure]

Ultimately, I do recommend publication, but provided the few points below are answered.

Major points:

1. As mentioned: the end of the paper can be improve. Specifically, while the beginning (the methods, mostly) is well documented, the last part (the results) appears too short. This creates a sort of frustration, as the reader realizes the emulator performs well but is not always sure what physical behavior/process is actually well emulated. A couple of sentences, here and there, to remind the reader of the main conclusion of already cited studies (e.g. Kravitz et al., 2015; Andrews et al., 2010) would help.

2. The paper lacks an introduction to EOFs! There is a quite lengthy explanation of what IRFs are and how they are obtained, but almost nothing about EOFs in the methods section. This should be re-balanced as EOFs are presented at the end of the paper. Maybe the part on IRFs could be shortened a little so as to avoid a too lengthy methods section.

3. The analysis of the performance of the emulators is limited to looking at some plots. It would be better to have at least a few quantitative metrics, to better understand the emulators' performance. Metrics could be provided in a table, both for the IRFs (time-series) and EOFs (spatial patterns).

4. This is more of a request, but it is maybe the most important point of my review. I believe the IRFs calculated by the authors should be provided as supplementary material. The paper would strongly benefit from it, as it would have much more impact on the modelers' community (and, therefore, it would be much more cited). This is especially true as the rationale behind the study is presented as being using those emulators in future studies of geoengineering scenarios. An Excel spreadsheet with one time-series per model and global variable should do it.

Minor points:

l. 3: I suggest adding "further" to "without relying *further* on GCMs" and removing "for every possible pathway".

l. 15: I find "be a more accurate estimate" than GCMs too strong. I would rather say "more cost-effective", especially as for GCMs the multi-model approach, as well as the multiple realizations, do compensate for the possible bias induced by natural variability. In the end, it is an issue of computation time requirement, not of accuracy.

l. 21: I don't like the word "interpolation" here.

l. 22: Change "fidelity" for something like: "spatial and temporal resolution".

l. 29: Define GeoMIP and explain briefly.

l. 38: Other variables such as precipitations are not always assumed to be strictly proportional to global mean temperature by simple models. E.g. some simple models use the relationship to GMT and RF by Andrews et al. (2010) for precipitations. Overall, I suggest being slightly less categorical.

l. 46-48: That sentence referring to Cao et al. (2015) should either be developed or removed. I found it incomprehensible.

l. 71: I suggest removing NPP of that study. See point below about figure S5.

l. 93-94: I find that last sentence too brief: please develop.

l. 113-114: I think a reminder that when the difference is done between these two simulations, you're assuming the system is linear.

l. 144-150: The drawback of training over lower forcings would be a reduced domain of validity of the emulators, wouldn't it?

l. 191-193: This drifting issue makes one wonder about the results of the study... Maybe this should be slightly expended. Can the drift be actually explained? How significant is it?

[Figure]

l. 202: Example of where one or two sentences could improve the paper. Explain/recall why there is a difference in the fast response.

l. 230-233: That sentence is a bit obscure. Is this a property of the IRFs or the GCMs? Develop.

l. 234: Change "indicate" to "provide"?

l. 239-242: I honestly don't understand how the authors can claim that there is "no evidence of non-linearity". What would be the evidence? Do you mean that the non-linearity is negligible, and therefore captured by the IRF?

l. 242-245: As in the abstract, I find this statement far too strong. It should be moderated. I would basically remove the sentence, unless actual proof can be provided...

l. 256: Change "metrics" to "impacts"?

l. 260: Again, moderate a little bit: more insight *on some aspects*. Maybe recall the computing-efficiency of the emulators. I believe this is definitely their most significant strength.

l. 267: I fear the use of the word "moments" here may be confusing for the majority of the community. Maybe write "*statistical* moments", or expend or rephrase.

l. 281: It is always possible to develop emulators, except that they have to be non-linear. So basically, the next step is to build box models with non-linear coefficients.

Fig.1: Check units. For this specific plot, the IRF units should be e.g. °C/[W/m2] I think. Check also units for precipitations.

Fig.3: Units.

Fig.5: Needs a title over each map.

Fig. S1: Could be in main text.

Fig. S5: Units are likely wrong. NPP should be tens of PgC/yr. But more importantly I

suggest removing that plot on NPP. NPP is not a variable of the climate system stricto sensu, it is a variable of the carbon-cycle. NPP responds firstly to changes in atmospheric CO2, then to changes in climate and incoming radiation (at least in current-generation ESMs). The response to CO2 is strongly non-linear in intensity (can be captured with a simple log function, at global scale) and it is virtually instantaneous at the yearly time-scale. So here there is virtually no difference between the two simulations because NPP is basically responding to the annual atmospheric CO2. In short: the IRF approach is *not* the right approach for NPP: wrong driving variables, and wrong time-scale.
* * *

---

## Author Response (AR1)

**Response to Anonymous Referee #1**

Thanks again for your careful review; our responses are in blue below. We uploaded a response shortly after this review, but waited until the other reviews were complete before revising the paper; this comment is mostly redundant with our original response, but is now also updated based on the revisions made.

The authors propose a nonparametric emulator aimed at reproducing geoengineering scenarios. Using dynamical linear models, they propose a formulation of the emulator as a convolution of the forcing and the impulse responses, and test this approach for two geoMIP scenarios for some variables of interests.

The manuscript is overall well written and presents an interesting problem, but I believe that in its present form is not suitable for publication and, in order to be reconsidered, needs to be considerably improved in many parts. The proposed method would have considerable limitations if it is to be expanded beyond the narrow context of this work, e.g. annual averages for two model runs. Further, the validation setting is extremely limited, not based on any metric, and completely ignores the emulation uncertainty.

We agree that quantifying emulator accuracy (i.e., evaluating based on specific metrics) is essential; Table 1 now includes a comparison for each model and each variable of the root-mean-square difference between the emulator prediction and the simulation, compared with the natural variability.

The description in the paper of how to extend results to cover sub-annual time-scales was poorly worded and thus quite misleading. The case of evaluating sub-annual variables in response to annual-mean forcing is already covered by the method. Indeed we considered one such variable but did not include a plot in the original paper,. It has now been included both because it makes this point, and because it also illustrates a breakdown of linearity (these two are unrelated). The description in the paper regarding sub-annual scales was intended to refer to the case where the forcing varies at sub-annual time-scales. Conceptually this is a trivial extension, but would require additional training data; this is not a limitation of our method per se, but only a limitation that information cannot be invented from nothing. (You cannot infer the response to seasonally-varying forcing from a simulation where the forcing was constant, no matter what method you apply.)

Also note that our formalism for capturing the linear forced response is intentionally provided in a more general fashion than most previous research. The key distinction relative to the existing literature aimed at the similar problem for non-geoengineering forcing is that we only assume linearity, and do not choose to make additional ad hoc (and apparently unnecessary) assumptions regarding the form of the dynamics; in doing so it helps clarify the additional assumptions made elsewhere. We clarified this in the introduction to better articulate the relationship with existing approaches.

We disagree regarding the limitation of the validation setting, as described in more detail below. The perceived limitation regarding annual averages is not a limitation but simply poor wording on our part. Validation on one forcing scenario is sufficient to demonstrate that linearity is a sufficiently useful approximation for the variables that we consider here. Since that is the only assumption we make, additional validation scenarios would not add any further value. (If the reviewer is used to dealing with nonlinear emulators, for example as used in model tuning, then it would be absolutely true that you can neither tune nor validate such an emulator from single simulations.)

**General comments**
• The validation setting is extremely limited: the proposed approach is fit for the G1 scenario, and then used to extrapolate G2. Also, for the G1 scenario the emulator is likely to work well, since it consists of impulse functions. A considerable amount of work is needed to perform more tests under different forcing scenarios. While the geoMIP is limited in size, the CMIP5 or other large multi-model ensembles could be used to validate the forcing part of this emulator.

We demonstrated that a *linear* emulator trained on one scenario successfully follows the behaviour of another. Since linearity is the only assumption we make, then (i) the emulator can be uniquely specified from a single forcing simulation such as G1, and (ii) no value would be added by validating it on additional forcing scenarios. By demonstrating that linearity holds for forcing levels up to 4xCO2

and for levels of solar reduction sufficient to compensate this, it will clearly also hold for *any* other smaller-amplitude forcing scenario.  We reworded the paper to clarify this.

• The present version puts very little emphasis of the uncertainty of the scenario estimation. The validation essentially consists in eyeballing many plots of the emulator against the original computer model, with no attempt to quantify the fit or, most importantly, to assess how the internal variability of the model is reproduced by the emulator. The definition itself of 'climate variability' $n_i(t)$ is unclear.  Are the authors assuming a white noise? Also, I would assume that this noise is independent for different variables, but it should be clearly stated.

We added a quantification of the fit; we agree that not doing so was an oversight in the original manuscript.

We also clarify that the emulator is intended to only capture the forced response, and is not intended to reproduce internal variability.  Insofar as we are only interested in capturing the forced response, we do not need to make any assumptions about the nature of the climate variability as it enters only as "noise" in our ability to estimate the forced response.  This has been clarified in the manuscript.

• This approach will have significant limitations at finer temporal scales. The authors briefly discuss this when they mention how we can impose $h = h(\_,m)$.  This solution is not straightforward, as a nonparametric estimation of 12 different impulse responses will require more scenarios (surely more than two) to have reliable estimates. The authors somewhat acknowledge it when they state that additional simulations would be required, but in an off-the-shelf ensemble such as geoMIP, where no more scenarios are readily available, this is a strong limit of this approach. This will be become even more evident for finer temporal scales, e.g. weekly or daily data.

As noted above, we apologize for badly worded text here that was misleading.  To clarify, if the *forcing* does not vary significantly over the course of the year, then emulating GCM response at finer temporal scales is not intrinsically more difficult for this or any other emulator, although the signal to noise ratio (SNR) will likely be poorer (a limitation that is intrinsic to the information contained in the training data, and has nothing to do with the method itself).

If the intent is to capture the response to seasonally-varying forcing, then unless arbitrary assumptions are made regarding the seasonal dependence of the impulse response, one would need at least as many independent forcing scenarios as degrees of freedom of the seasonal response (i.e., 12 if one wants to distinguish how the response depends on monthly-varying forcing).  This is not a limitation of the formulation we use (it would be a trivial extension), rather it is an intrinsic limitation on the knowledge of the response that holds for any such approach.  Of course, for evaluating climate change in response to different pathways of greenhouse gas forcing and solar geoengineering, the forcing varies only slowly from year to year, so that the additional training data is not needed.

We did a poor job of articulating the distinction between these two cases, and have corrected this.

We focused on information about annual-average behaviour because it is indeed useful, both for geoengineering and more general climate science applications.  We have added one sub-annual variable in revision; the annual-minimum sea ice extent.  This is provided not to illustrate the ability to project sub-annual variables, but because it illustrates a case where nonlinearity is significant and the emulator does not perform well.  (That this particular variable happens to be nonlinear is unrelated to the fact that this particular variable is at a sub-annual time scale.)

• The results and the discussion do not mention model differences, and most importantly what do they mean. Does the emulator estimate different impulse responses for different models? I would expect so, and I would expect these differences to convey information on how the models differ. For example, HadCM3 and HadGEM2-ES will likely display similar responses as both models are released from the Hadley Centre.

Our intent was to develop an approach for emulating climate models, not for describing the differences between them, for which there is already an abundant literature.  We added some text to this effect and appropriate references.

• The part on grid-scale emulation must be extended. Firstly, the methodology is unclear: a clear explanation of how were the EOFs selected must be presented, either in the main text or in the supplement. Secondly, as before, a more formal assessment of the pattern similarity is needed, as eyeballing figure 5 is not enough to convince that the emulator is performing well.

> We agree that this section was too terse. We added both a more complete description, and a more formal pattern similarity assessment (rms differences).

**Specific comments**
• Title: what the authors present is not a multi-model emulator, in the sense that it independently fits each model and does not assume interdependencies.

> We agree that there are multiple ways of interpreting the phrase "multi-model". We meant it only in the sense that the end result is a set of emulators (i.e., in the same sense that CMIP or GeoMIP are "multi-model" ensembles.) This is a not uncommon usage of the adjective, nonetheless we removed it from the title (shorter is better).

• pag. 1 l.16-17. The claim that the 'emulator prediction may be a more accurate estimate [...] of the models' response than an actual simulation' is very questionable. The emulator is not meant to replace a climate model, it's just a faster approximation that is used to explore the input space in a computationally efficient manner. While emulators are arguably a useful tool for calibration and, as in this case, scenario extrapolation, they cannot replace the physics of the climate model and they are useful only as long as the training set from the climate model is meaningful.

> This point was not well worded in the original manuscript, and we have endeavoured to clarify what we meant, both here and in the text.

> And, of course the climate model is needed to generate training data for the emulator, and does not replace climate models. We also agree that the emulator is only useful so long as the training set is meaningful.

> As to whether the emulator prediction is a more accurate estimate for some specific scenario, that depends on the purpose. If the goal is to estimate the *forced* component of the response, isolated from natural variability, then it may well be true that simulating at a higher forcing amplitude, to give a higher SNR, and then scaling the response, would indeed give a better estimate *for a given amount of computation*. (If the system were perfectly linear in its response to forcing, this is self-evident.) If computational power is unlimited, then of course the best answer for the forced response would come from a sufficiently large ensemble of GCM simulations of the specific scenario.

> Fundamentally one is simply trading off the uncertainty in the forced response that comes from superimposed natural variability from the uncertainty that comes from nonlinearity. Given sufficient computation to conduct only one single simulation, then it is not a priori clear whether the best estimate of the forced response in a particular scenario is obtained by simulating that particular scenario, or simulating a higher SNR scenario and using an emulator to "scale" it.

• pag 1. l.19-20. Actually, emulators are much more popular in model calibration and local sensitivity analysis of physical parameters then in projections of anthropogenic forcings. Only very recently this methodology have been extended to deal with forcings. This introductory part must be rewritten with a more extensive literature review on traditional emulators.

> We added a comment and reference to acknowledge the breadth of application of emulators. However, it is only the use of emulators to deal with forcings that is directly relevant to the case here. (As a side note, 1990 probably doesn't count as "very recently" any more!)

• pag. 4, eq (1) and onwards. It is somewhat inappropriate to represent the emulator as a convolution given that the authors are effectively using just annual averages. A reformulation in terms of discrete sums is necessary.

> Agreed. The emulator is now given in terms of discrete sums; the continuous-time is still introduced in case some readers are more comfortable with it, and in particular, we motivate the influence of the

climate variability spectrum through Laplace transforms of the continuous-time convolution equation; readers are undoubtedly more comfortable with them than Z-transforms that would otherwise be needed.

• pag. 4, line 101. h(_ ) was never defined.
• pag. 6, line 161. Poor choice of pedix in $f_t(t)$, please reformulate.

Thanks; fixed.

• Figures. What is the unit measure of precipitation? Also, are the all figures expressed as anomaly with respect to a reference value? If so, what is it?

Oops; fixed. Sorry, final version of figures were generated but didn't get included! Not sure how that happened or passed final proof-reading. Units are in mm/day, and are all in anomalies with respect to the preindustrial control values.

**Response to Anonymous Referee #2**

Thanks for the comments; our responses are in blue below.

The authors used model results from Geoengineering Model Intercomparison Project (GeoMIP) to test the linearity of the climate response to external forcings. The authors first constructed a climate emulator based on a convolution of impulse response function using results from GeoMIP G1 simulations involving abrupt changes in atmospheric CO2 and solar irradiance. Then the authors used the climate emulator to predict climate consequences of the GeoMIP G2 simulations involving gradual change in atmospheric CO2 and solar irradiance. For climate variables including temperature, precipitation, and annual mean Northern Hemisphere sea ice extent, the emulator does a good job in reproducing climate model simulated temporal evolution and spatial distribution.

The use of impulse response function to emulate climate model results is not new. The novelty of this study is that it extends the application of impulse response function to the simulations involving both CO2 and solar forcing. This extension advances our understanding of climate response to external forcing, and in particular, climate response to solar geoengineering. The ms is well written. I recommend publication after the following issues are addressed:

1. The GeoMIP simulations are limited to a period of 50 years. Over longer timescales (several centuries), response from deep ocean dynamics would become important. Many aspects of ocean dynamics response (e.g., thermohaline circulation) are nonlinear. So the question is: To what extend the linear emulator would be valid in reproducing long-term climate response involving feedbacks from deep ocean dynamics?

> Agreed; we have added a comment to the manuscript regarding this point, including both a citation to the literature on AMOC nonlinearity (acknowledging the limitation in using 50-year training simulations), and to one study indicating that the net response (combining CO2 forcing and solar reduction) does not drift (which at least gives some confidence that the long-term climate response would not be radically different from the short-term, at least in one model).

2. A large part of the residual response of the hydrological cycle over land to solar geoengineering is due to the direct effect of increasing atmospheric CO2 on vegetation (stomatal, leaf area index, etc.), which cannot be offset by reduced solar forcing. Assumedly, this part of hydrological cycle response is nonlinear. This issue should be discussed.

> The reviewer raises an important distinction here, between whether the overall processes involved are nonlinear, versus whether the perturbation in the response is approximately proportional to a perturbation in the forcing (so double the forcing doubles the perturbation).  As long as the nonlinear relationships in question are differentiable at the current equilibrium point, then by definition there is a linear first-order response, although the size of perturbation for which that is relevant is not a priori clear.  We have endeavoured to clarify the wording regarding linearity in a few places, both at the beginning of section 2, and with a more thorough discussion of which variables have an apparently nonlinear response in which simulations (since the difference between emulated and simulated responses indicates nonlinearity, if for example G2 precipitation (suppressing the slow response) is well predicted but not the 1%CO2 simulation, then one can conclude that the fast response is relatively linear, but that nonlinearities arise in the slow response to precipitation.)

3. The method used to emulate spatial pattern of temperature and precipitation is not clear. How EOFs were constructed, selected, and applied to generate the spatial pattern of climate change? These should be elaborated.

Agreed; we have added section 2b to describe the spatial EOF analysis.

**Response to Anonymous Referee #3**

Thanks for the detailed review! This is very helpful; our comments are in blue below.

The paper entitled "Multi-model dynamic climate emulator for solar geoengineering" by MacMartin & Kravitz presents a simple numerical emulator of the complex GeoMIP models which could be used to discuss Geoengineering scenarios. This paper is well written, fairly straightforward, and is interesting – I believe – for the community.

One could wonder, however, if ACP is the best journal for publication, as a lot of technical detail regarding the modeling (i.e. establishing the response functions) is given, whereas the more physical aspects remain (maybe too) brief. Maybe GMD would have been a better choice. But that is ultimately an editorial issue. And I don't think this point alone prevents publication in ACP, especially as the physics is well understood and already published elsewhere. It goes in favor, however, of improving the narrative so that the reader can grasp both the modeling approach and the modeled physical processes.

We have added some text regarding physical processes (particularly with regards to fast and slow responses, and some comments regarding what the difference between observed nonlinear effects between G2 and 1%CO2 simulations implies about nonlinearities in fast and slow responses).

Ultimately, I do recommend publication, but provided the few points below are answered.

Major points:

1. As mentioned: the end of the paper can be improve. Specifically, while the beginning (the methods, mostly) is well documented, the last part (the results) appears too short. This creates a sort of frustration, as the reader realizes the emulator performs well but is not always sure what physical behavior/process is actually well emulated. A couple of sentences, here and there, to remind the reader of the main conclusion of already cited studies (e.g. Kravitz et al., 2015; Andrews et al., 2010) would help.

We agree with this comment and have added both references to previous literature and a more detailed description of the results and their physical interpretation.

2. The paper lacks an introduction to EOFs! There is a quite lengthy explanation of what IRFs are and how they are obtained, but almost nothing about EOFs in the methods section. This should be re-balanced as EOFs are presented at the end of the paper. Maybe the part on IRFs could be shortened a little so as to avoid a too lengthy methods section.

Agreed; we have added section 2b in the methods to discuss EOFs; insofar as EOF analysis is standard (in contrast to IRFs) in climate science, this section is shorter. As the IRFs are crucial to the paper, we have not shortened.

3. The analysis of the performance of the emulators is limited to looking at some plots. It would be better to have at least a few quantitative metrics, to better understand the emulators' performance. Metrics could be provided in a table, both for the IRFs (timeseries) and EOFs (spatial patterns).

> Agreed; we have added a table with calculation of the root-mean-square deviation between predicted and simulated results for each model and each variable, compared with the rms of climate variability.

4. This is more of a request, but it is maybe the most important point of my review. I believe the IRFs calculated by the authors should be provided as supplementary material. The paper would strongly benefit from it, as it would have much more impact on the modelers' community (and, therefore, it would be much more cited). This is especially true as the rationale behind the study is presented as being using those emulators in future studies of geoengineering scenarios. An Excel spreadsheet with one time-series per model and global variable should do it.

> An excellent suggestion; we have included these in supplementary material.

Minor points:

l. 3: I suggest adding "further" to "without relying *further* on GCMs" and removing "for every possible pathway".

> Good! Done.

l. 15: I find "be a more accurate estimate" than GCMs too strong. I would rather say "more cost-effective", especially as for GCMs the multi-model approach, as well as the multiple realizations, do compensate for the possible bias induced by natural variability. In the end, it is an issue of computation time requirement, not of accuracy.

> This sentence has been deleted from the abstract, as putting in the appropriate caveats is too long for an abstract; we add the extra phrasing later in the text. We agree that if there was no computation time limit, then the GCM would be more accurate. However, given a finite amount of computation time, then it *is* possible that the emulator is indeed more accurate; this is too subtle for the abstract.

l. 21: I don't like the word "interpolation" here.

> Agreed, changed.

l. 22: Change "fidelity" for something like: "spatial and temporal resolution".

> The emulator in principle is capable of the same spatial and temporal resolution as the GCM, and even debating their relative accuracy as predictors of the forced-response would require a lengthy discussion. We haven't come up with a better word that is an accurate description of the advantages of the GCM.

l. 29: Define GeoMIP and explain briefly.

> The definition and explanation are expanded towards the end of this section when the GeoMIP simulations are being explicitly referred to (this also benefits from your suggestion of moving figure S1 into the main text).

l. 38: Other variables such as precipitations are not always assumed to be strictly proportional to global mean temperature by simple models. E.g. some simple models use the relationship to GMT and RF by Andrews et al. (2010) for precipitations. Overall, I suggest being slightly less categorical.

> Thanks for correcting our error; reworded.

l. 46-48: That sentence referring to Cao et al. (2015) should either be developed or removed. I found it incomprehensible.

> Thanks – completely reworded to clarify.

l. 71: I suggest removing NPP of that study. See point below about figure S5.

> Agreed, good point.

l. 93-94: I find that last sentence too brief: please develop.

> We moved the important aspects of the sentence further down after the equations, where it is better motivated.

l. 113-114: I think a reminder that when the difference is done between these two simulations, you're assuming the system is linear.

> Good call!

l. 144-150: The drawback of training over lower forcings would be a reduced domain of validity of the emulators, wouldn't it?

> Not necessarily.  Basically it's a trade-off between signal-to-noise ratio and nonlinearity; we reworded this paragraph to clarify.

l. 191-193: This drifting issue makes one wonder about the results of the study: : : Maybe this should be slightly expended. Can the drift be actually explained? How significant is it?

> We are quite certain that the drift is due to initialization (that is, a few of the models were not fully spun up and continued to drift).  If the starting conditions had been well documented, so that we could download the control run, this could be fixed; unfortunately this is not true and so we simply discarded those models where the drift was significant.  (As a result, this is no longer an issue for this study.)

l. 202: Example of where one or two sentences could improve the paper. Explain/recall why there is a difference in the fast response.

> Done!

l. 230-233: That sentence is a bit obscure. Is this a property of the IRFs or the GCMs? Develop.

> Yeah, that was a pretty badly worded sentence.  We clarified.

l. 234: Change "indicate" to "provide"?

> Agreed!

l. 239-242: I honestly don't understand how the authors can claim that there is "no evidence of non-linearity". What would be the evidence? Do you mean that the nonlinearity is negligible, and therefore captured by the IRF?

> Sentence is completely reworded in order to clarify.  (If the nonlinearity was non-negligible, then the emulator based off a linear assumption and trained at one forcing

level would not have matched the response to a different forcing. We therefore conclude that nonlinearity is not too large, at least for these variables.)

l. 242-245: As in the abstract, I find this statement far too strong. It should be moderated. I would basically remove the sentence, unless actual proof can be provided: : :

The paragraph is reworded to better clarify exactly what we meant. Note that the observation is trivially true if indeed the model were perfectly linear; in general there is a trade-off between uncertainty introduced from natural variability, and errors introduced due to nonlinearity. Without conducting a large ensemble of G2 to provide a "truth" for the forced-response, it is not possible to separate these two errors and determine whether the emulated response actually does provide a better estimate of the forced response, or whether it is simply possible that it does (and again, the potential is trivially true, and that is the only statement made here, so we disagree that the statement is "too strong" though agree that it was badly worded!)

l. 256: Change "metrics" to "impacts"?

The word "impacts" is often associated with a specific meaning in some of the climate change literature, as the actual things humans care about; while there is often overlap with climate variables in a GCM, there might not be (e.g. vector-borne diseases), and so we consciously avoided the term "impacts".

l. 260: Again, moderate a little bit: more insight *on some aspects*. Maybe recall the computing-efficiency of the emulators. I believe this is definitely their most significant strength.

Agreed, changed.

l. 267: I fear the use of the word "moments" here may be confusing for the majority of the community. Maybe write "*statistical* moments", or expend or rephrase.

On reflection, including "statistical moments" was unnecessary to convey the point; we condensed to simply refer to extremes.

l. 281: It is always possible to develop emulators, except that they have to be nonlinear. So basically, the next step is to build box models with non-linear coefficients.

Reworded to clarify that one can always develop nonlinear emulators, although they would require either a priori assumptions or multiple forcing scenarios for training.

Fig.1: Check units. For this specific plot, the IRF units should be e.g. _C/[W/m2] I think. Check also units for precipitations.

Actually, not quite; the units here should be degrees C for a 4xCO2 (which is not the same radiative forcing in every model). We added this to the caption.

Fig.3: Units.

Thanks! (Oops.)

Fig.5: Needs a title over each map.

Thanks, done.

Fig. S1: Could be in main text.

Agreed, moved.

Fig. S5: Units are likely wrong. NPP should be tens of PgC/yr. But more importantly I suggest removing that plot on NPP. NPP is not a variable of the climate system stricto sensu, it is a variable of the carbon-cycle. NPP responds firstly to changes in atmospheric $CO_2$, then to changes in climate and incoming radiation (at least in currentgeneration ESMs). The response to $CO_2$ is strongly non-linear in intensity (can be captured with a simple log function, at global scale) and it is virtually instantaneous at the yearly time-scale. So here there is virtually no difference between the two simulations because NPP is basically responding to the annual atmospheric $CO_2$. In short: the IRF approach is *not* the right approach for NPP: wrong driving variables, and wrong time-scale.

Agreed, NPP removed.

[revised manuscript text omitted]